# SODA10M: A Large-Scale 2D Self/Semi-Supervised Object Detection Dataset for Autonomous Driving

**Jianhua Han**[1*], **Xiwen Liang**[2*], **Hang Xu**[1†], **Kai Chen**[4], **Lanqing Hong**[1], **Jiageng Mao**[3],

**Chaoqiang Ye**[1], **Wei Zhang**[1], **Zhenguo Li**[1], **Xiaodan Liang**[2†], **Chunjing Xu**[1]

## Abstract

Aiming at facilitating a real-world, ever-evolving and scalable autonomous driving system, we present a large-scale dataset for standardizing the evaluation of different self-supervised and semi-supervised approaches by learning from raw data, which is the first and largest dataset to date. Existing autonomous driving systems heavily rely on 'perfect' visual perception models (*i.e.*, detection) trained using extensive annotated data to ensure safety. However, it is unrealistic to elaborately label instances of all scenarios and circumstances (*i.e.*, night, extreme weather, cities) when deploying a robust autonomous driving system. Motivated by recent advances of self-supervised and semi-supervised learning, a promising direction is to learn a robust detection model by collaboratively exploiting large-scale unlabeled data and few labeled data. Existing datasets (*i.e.*, BDD100K, Waymo) either provide only a small amount of data or covers limited domains with full annotation, hindering the exploration of large-scale pre-trained models. Here, we release a Large-Scale 2D **S**elf/semi-supervised **O**bject **D**etection dataset for **A**utonomous driving, named as **SODA10M**, containing 10 million unlabeled images and 20K images labeled with 6 representative object categories. To improve diversity, the images are collected within 27833 driving hours under different weather conditions, periods and location scenes of 32 different cities. We provide extensive experiments and deep analyses of existing popular self-supervised and semi-supervised approaches, and give some interesting findings in autonomous driving scope. Experiments show that SODA10M can serve as a promising pre-training dataset for different self-supervised learning methods, which gives superior performance when fine-tuning with different downstream tasks (*i.e.* detection, semantic/instance segmentation) in autonomous driving domain. This dataset has been used to hold the ICCV2021 SSLAD challenge. More information can refer to https://soda-2d.github.io.

## 1 Introduction

Autonomous driving technology has been significantly accelerated in recent years because of its great applicable potential in reducing accidents, saving human lives and improving efficiency. In a real-world autonomous driving system, object detection plays an essential role in robust visual perception in driving scenarios.

A major challenge of training a robust object detector for autonomous driving is how to effectively handle the rapid accumulation of unlabeled images. For example, an autonomous vehicle equipped

---

[1] Huawei Noah's Ark Lab    [2] Sun Yat-Sen University    [3] The Chinese University of Hong Kong
[4] Hong Kong University of Science and Technology    [*] These two authors contribute equally.
[†] Corresponding authors: xdliang328@gmail.com & xu.hang@huawei.com

35th Conference on Neural Information Processing Systems (NeurIPS 2021) Track on Datasets and Benchmarks.

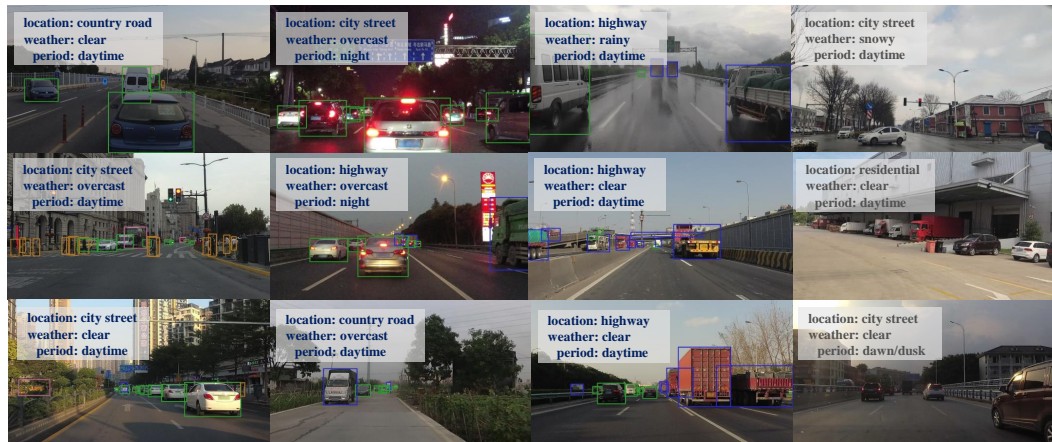

Figure 1: Examples of challenging environments in our SODA10M dataset, including 10 million images covering different weather conditions, periods and locations. The first three columns of images are from SODA10M labeled set, and the last column is from the unlabeled set.

with 5 camera sensors (same to [48]) can capture 288K images (2Hz) in 8 hours. However, exhaustively labeling bounding boxes on those images generally takes about 136 days with 10 experienced human annotators in the annotating pipeline. The speed gap between data acquisition and data annotation leads to an intensive demand of leveraging large-scale unlabeled scenes to boost the accuracy and robustness of detectors. To tackle this important problem, emerging approaches of semi-supervised [43, 40, 61, 30] and self-supervised learning [4, 19, 21, 5] shows great potentials and may become the next-generation industrial solution for training robust perception models.

To explore self/semi-supervised methods in the autonomous driving scenarios, researchers will meet two main obstacles: 1) Lacking suitable data source. Popular datasets (*i.e.*, ImageNet [7], YFCC [50]) contains mostly common images, and available autonomous driving datasets (*i.e.*, nuScenes [1], Waymo [48], BDD100K [60]) are not large enough in scale or diversity. 2) Lacking a benchmark tailored for autonomous driving. Current methods [21, 5] are mostly tested on common images, such as ImageNet [7]. A comprehensive evaluation of those methods in driving scenarios and new observations are strongly required (*i.e.*, which self-supervised method is more suitable when pre-training on autonomous driving datasets, or what is the difference compared with ImageNet [7]?).

To this end, we develop the first and largest-Scale 2D **S**elf/semi-supervised **O**bject **D**etection dataset for **A**utonomous driving (SODA10M) that contains 10 million road images. Aiming at self/semi-supervised learning, our SODA10M dataset can be distinguished from existing autonomous Driving datasets from three aspects, including *scale*, *diversity* and *generalization*.

***Scale***. As shown in Table 1, SODA10M is significantly larger than existing autonomous driving datasets like BDD100K [60] and Waymo [48]. It contains 10 million images of road scenes, which is ten times more than Waymo [48] even with a much longer collecting interval (ten seconds per frame). Considering the purpose of evaluating different self-supervised and semi-supervised learning methods, we use the number of images, instead of the labeled images, for comparison.

***Diversity***. SODA10M comprises images captured in 32 cities under different scenarios (*i.e.*, urban, rural) and circumstances (*i.e.*, night, rain, snow), while most present self-driving datasets [63, 48] are less diverse (*i.e.*, Waymo [48] doesn't contain snowy scene). Besides, the total driving time of SODA10M dataset is 27833 hrs covering four seasons, which is 25x, 5000x and 4300x longer than the existing BDD100K [60], nuScenes [1] and Waymo [48] dataset.

***Generalization***. The superior scale and diversity of the SODA10M dataset ensure great generalization ability as a pre-training dataset over all existing autonomous-driving datasets. Observed from evaluations of existing self-supervised algorithms, the representations learned from SODA10M unlabeled set are superior to that learned from other driving datasets like Waymo [48], ranking top in 9 out of 10 downstream detection and segmentation tasks (see Sec. 4.3 for more details).

Table 1: Comparison of dataset statistics with existing driving datasets. Night/Rain indicates whether the dataset has domain information related to night/rainy scenes. Video represents whether the dataset provides video format or detailed chronological information. Considering the purpose of evaluating different self/semi-supervised learning methods, we use the number of images, instead of the labeled images, for comparison.

| Dataset | Images | Cities | Night/Rain | Video | Driving hours | Resolution |
|---|---|---|---|---|---|---|
| Caltech Pedestrian [9] | 249K | 5 | ✗/✗ | ✓ | 10 | 640×480 |
| KITTI [14] | 15K | 1 | ✗/✗ | ✗ | 6 | 1242×375 |
| Citypersons [63] | 5K | 27 | ✗/✗ | ✗ | - | 2048×1024 |
| BDD100K [60] | 100K | 4 | ✓/✓ | ✓ | 1111 | 1280×720 |
| nuScenes [1] | 1.4M | 2 | ✓/✓ | ✓ | 5.5 | 1600×900 |
| Waymo Open [48] | 1M | 3 | ✓/✓ | ✓ | 6.4 | 1920×1280 |
| SODA10M (Ours) | **10M** | **32** | ✓/✓ | ✓ | **27833** | 1920×1080 |

We also provide experiments and in-depth analysis of prevailing self-supervised and semi-supervised approaches and some observations under autonomous driving scope. For example, dense contrastive method (*i.e.*, DenseCL [54]) performs better than global contrastive methods (*i.e.*, MoCo-v1 [21]) when pre-training on ImageNet [7] (39.9% vs. 39.0%, fine-tuning with object detection task on SODA10M labeled set). However, Dense contrastive method preforms worse when pre-training on autonomous driving dataset (38.1% vs. 38.9%) due to the reason that pixel-wise contrastive loss may not suitable for complex driving scenarios (refer to Sec. 4.3.2). More findings can be found in Sec. 4.

SODA10M dataset has been released and used to hold the ICCV2021 SSLAD challenge[1], which aims to investigate current ways of building next-generation industry-level autonomous driving systems by resorting to self/semi-supervised learning. Until now, the challenge already attracts more than 130 teams participating in and receive 500+ submissions.

To conclude, our main contributions are: 1) We introduce SODA10M dataset, which is the largest and most diverse 2D autonomous driving dataset up to now. 2) We introduce a benchmark of self/semi-supervised learning in autonomous driving scenarios and provide some interesting findings.

## 2 Related Work

**Driving datasets** have gained enormous attention due to the popularity of autonomous self-driving. Several datasets focus on detecting specific objects such as pedestrians [9, 63]. Cityscapes [6] provides instance segmentation on sampled frames, while BDD100K [60] is a diverse dataset under various weather conditions, time and scene types for multitask learning. For 3D tasks, KITTI Dataset [15, 14] was collected with multiple sensors, enabling 3D tasks such as 3D object detection and tracking. Waymo Open Dataset [48] provides large-scale annotated data with 2D and 3D bounding boxes, and nuScenes Dataset [1] provides rasterized maps of relevant areas.

**Supervised learning** methods for object detection can be roughly divided into single-stage and two-stage models. One-stage methods [35, 11, 37] directly outputs probabilities and bounding box coordinates for each coordinate in feature maps. On the other hand, two-stage methods [22, 44, 34] use a Region Proposal Network (RPN) to generate regions of interest, then each proposal is sent to obtain classification score and bounding-box regression offsets. By adding a sequence of heads trained with increasing IoU thresholds, Cascade RCNN [2] significantly improves detection performance. With the popularity of the vision transformer, more and more transformer-based object detectors [53, 39] have been proposed to learn more semantic visual concepts with larger receptive fields.

**Self-supervised learning** approaches can be mainly divided into pretext tasks [8, 62, 42, 41] and contrastive learning [21, 5, 4, 19]. Pretext tasks often adopt reconstruction-based loss functions [8, 42, 17] to learn visual representation, while contrastive learning is supposed to pull apart negative pairs and minimize distances between positive pairs, achieved by training objectives such as InfoNCE [51]. MoCo [21, 5] constructs a queue with a large number of negative samples and a moving-averaged encoder, while SimCLR [4] explores the composition of augmentations and the effectiveness of non-linear MLP heads. SwAV [3] introduces cluster assignment and swapped prediction to be more robust about false negatives, and BYOL [19] demonstrates that negative samples are not prerequisite

---
[1]https://competitions.codalab.org/competitions/33288

to learn meaningful visual representation. For video representation learning, early methods are based on input reconstruction [24, 25, 31, 32], while others define different pretext tasks to perform self-supervision, such as frame order prediction [33], future prediction [47, 52] and spatial-temporal jigsaw [29]. More recently, contrastive learning is integrated to learn temporal changes [18, 59].

**Semi-supervised learning** methods mainly consist of self training [58, 57] and consistency regularization [45, 61, 20]. Consistency regularization tries to guide models to generate consistent predictions between original and augmented inputs. In the field of object detection, previous works focus on training detectors with a combination of labeled, weaky-labeled or unlabeled data [26, 49, 13], while recent works [27, 46] train detectors with a small set of labeled data and a larger amount of unlabeled images. STAC [46] pre-trains the object detector with labeled data and generate pseudo labels on unlabeled data, which are used to fine-tune the pre-trained model. Unbiased Teacher [38] further improves the process of generating pseudo labels via teacher-student mutual learning.

## 3 SODA10M

We collect and release a large-scale 2D dataset SODA10M to promote the development of self-supervised and semi-supervised learning approaches for lifting autonomous driving system into more real-world scenarios. Our SODA10M contains 10M unlabeled images and 20K labeled images, which is split into training(5K), validation(5K) and testing(10K) sets.

### 3.1 Data Collection

**Collection.** The image collection task is distributed to the tens of thousands of taxi drivers in different cities by crowdsourcing. They need to use a mobile phone or driving recorder (with high resolution 1080P+) to obtain images. To achieve better diversity, the images are collected every 10 seconds per frame and obtained in diverse weather conditions, periods, locations and cities.

**Sensor Specifications.** The height of the camera should be in the range of 1.4m-1.5m from the ground, depending on the different crowdsourced vehicles. Vehicles used to collect images are mostly passenger cars, e.g., sedan or van. The camera should be mounted on the rear mirror in the car, facing straight ahead. Besides, horizon needs to be kept at the center of the image, and the occlusion inside the car should not exceed 15% of the whole picture. Detailed camera settings, including distortion, exposure, white balance and video resolution, are given through the participants' instructions to each taxi driver in crowdsourcing system. Full instructions are shown in Appendix A.

**Quality Control.** We conduct the pre-collection and post-collection quality control to guarantee the high quality of the SODA10M dataset. The pre-collection quality control includes checking the camera position and imaging quality on each taxi. The post-collection quality control includes manual verification and those images of low quality (unclear imaging, strong reflection and incorrect camera position) will be returned for rectification.

**Data Distribution.** The images are collected mostly based on the pre-defined distribution of different cities and periods, while the other characteristics (e.g., weather, location) follows nature distribution. The total number of original collected images is 100M, and then 10M images with relatively uniform distribution are sampled out of these 100M images to get constructed into SODA10M dataset.

**Data Protection.** The driving scenes are collected in permitted areas. We comply with the local regulations and avoid releasing any localization information, including GPS and cartographic information. For privacy protection, we actively detect any object on each image that may contain personal information, such as human faces and license plates, with a high recall rate. Then, we blur those detected objects to ensure that no personal information is disclosed. Detailed licenses, terms of use and privacy are listed in Appendix A.

### 3.2 Data Annotation

Image tags (*i.e.*, weather conditions, location scenes, periods) for all images and 2D bounding boxes for labeled parts should be annotated for SODA10M. To ensure high quality and efficiency, the whole annotation process is divided into three steps: pre-annotation, annotation and examination.

**Pre-annotation**: In order to ensure efficiency, a multi-task detection model, which is based on Faster RCNN [44] and searched backbone [28], is trained on millions of Human-Vehicle images with bounding-box annotation and generate coarse labels for each image first. **Annotation**: Based on pre-annotated labels, annotators keep the accurate ones and correct the inaccurate labels. Each image is distributed to different annotators, and the images with the same annotation will be passed to the following process; otherwise, they would be distributed again. All annotators must participate in several courses and pass the examination for standard labeling. **Examination**: Senior annotators with rich annotation experience will review the image annotations in the second step, and the missing or incorrectly labeled images will be sent back for re-labeling. We exhaustively annotated car, truck, pedestrian, tram, cyclist and tricycle with tightly-fitting 2D bounding boxes in 20K images. The bounding box label is encoded as $(x, y, w, h)$, where $x$ and $y$ represent the top-left pixel of the box, and $w$ and $h$ represent the width and length of the box.

### 3.3 Data Statistics

**Labeled Set.** The labeled set contains 20K images with complete annotation. We carefully select 5K training set, 5K validation set, 10K testing set with disjoint sequence id (same sequence id denotes the corresponding images are taken by same car on same day). In order to study the influence of different self/semi-supervised methods on domain adaptation problem, the training set only contains images obtained in city streets of Shanghai with clear weather in the daytime, while the validation and testing sets have three weather conditions, locations, cities and two different periods of the day. The detailed analysis of SODA10M labeled set is shown in Appendix D.

**Unlabeled Set.** The unlabeled set contains 10M images with diverse attributes. As shown in Fig. 2(a), the unlabeled images are collected among 32 cities, covering a large part of eastern China. Compared with the labeled set, the unlabeled set contains not only more cities but also additional location (residential), weather (snowy) and period (dawn/dusk), according to the gray part in Fig. 2(b), Fig. 2(c) and Fig. 2(d). The rich diversity in SODA10M unlabeled set ensures the generalization ability to transfer to other downstream autonomous driving tasks as a pre-training or self-training dataset.

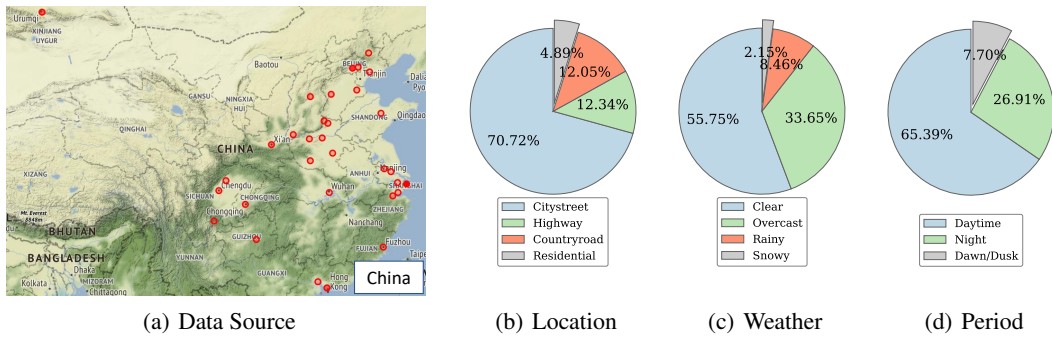

(a) Data Source     (b) Location     (c) Weather     (d) Period

Figure 2: Statistics of the unlabeled set. (a) Geographical distribution of our data sources. SODA10M is collected from 32 cities, and darker color indicates greater quantity. (b) Number of images in each location. (c) Number of images in each weather condition. (d) Number of images in each period.

**Clarification.** SODA10M contains a small number of labeled images compared with unlabeled images. However, we argue that the amount of labeled images is sufficient for two reasons: Firstly, SODA10M focuses on benchmarking self-supervised and semi-supervised 2D object detection methods, which requires SODA10M contain massive unlabeled images and a small number of labeled images for evaluation. Note that the purpose of SODA10M is not building a supervised benchmark that contains more annotations than the current datasets. Secondly, the scale of SODA10M labeled set is about the same as PASCAL VOC [12], which is considered as a popular-used self/semi-supervised downstream object detection dataset. Thus we believe that SODA10M provides sufficient labeled samples to evaluate different self/semi-supervised learning methods.

## 3.4 Comparison with Existing Datasets

We compare the SODA10M with other large-scale autonomous driving datasets (including BDD100K [60], nuScenes [1] and Waymo [48]) in the field of scale, driving time, collecting cities, driving conditions and fine-tuning results when regarded as upstream pre-training dataset.

Firstly, observed from Table 1, the number of images, driving time and collecting cities of SODA10M is 10M, 27833 hrs and 32 respectively, which is much larger than the current datasets BDD100K [60] (0.1M, 1111 hrs, 4 cities), nuScenes [1] (1.4M, 5.5 hrs, 2 cities) and Waymo [48] (1M, 6.4 hrs, 3 cities). Secondly, as shown in the driving conditions comparison results in Appendix D, our SODA10M is more diverse in driving conditions compared with nuSceness [1] and Waymo [48], and achieves competitive results with BDD100K [60]. Finally, with above characteristics, SODA10M achieves better generalization ability and obtains best performance compared with the other three datasets in almost all (9/10) downstream detection and segmentation tasks when regarded as the upstream pre-training dataset (refer to Sec. 4.3).

## 4 Benchmark

As SODA10M is regarded as a new autonomous driving dataset, we provide the fully supervised baseline results based on several representative one-stage and two-stage detectors. With the massive amount of unlabeled data, we carefully select most representative self-supervised and semi-supervised methods (Fig. 3) and study the generalization ability of those methods on SODA10M, and further provide some interesting findings under autonomous driving scope. To make the experiments easily reproducible, the code of all used methods has been open-sourced, and detailed experiment settings and training time comparisons are provided in Appendix B.

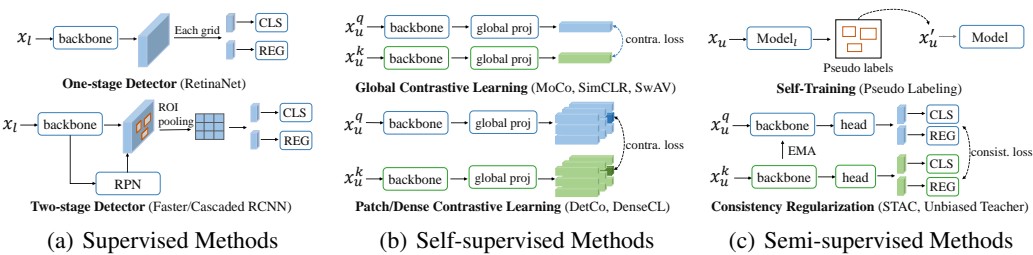

| (a) Supervised Methods | (b) Self-supervised Methods | (c) Semi-supervised Methods |

Figure 3: Overview of different methods used for building SODA10M benchmark. $X_l$ and $X_u$ denote for labeled and unlabeled set. $q$, $k$ represent for different data augmentations. For semi-supervised learning, the labeled set is also involved in training progress with supervised loss.

### 4.1 Basic Settings

We utilize Detectron2 [55] as our codebase for the following experiments. Following the default settings in Detectron2, we train detectors with 8 Tesla V100 with a batch size 16. For the 1x schedule, the learning rate is set to 0.02, decreased by a factor of 10 at 8th, 11th epoch of total 12 epochs, while 2x indicates 24 epochs. Multi-scale training and SyncBN are adopted in the training process and precise-BN is used during the testing process. The image size in the testing process is set to $1920 \times 1080$. Unless specified, the algorithms are tested on the validation set of SODA10M. COCO API [36] is adopted to evaluate the detection performance for all categories.

### 4.2 Preliminary Supervised Results

As shown in Table 2, the detection results of three popular object detectors (RetinaNet [35], Faster RCNN [44], Cascade RCNN [2] ) are compared. We observe that in the 1x schedule, Faster RCNN [44] exceeds RetinaNet [35] in mAP by 5.3% with a larger number of parameters, which is consistent with the traditional difference of single-stage and two-stage detectors. Equipped with a

stronger ROI-head, Cascaded RCNN [2] can further surpass Faster RCNN [44] by a large margin (3.9%). Training with a longer schedule can further improve the performance.

Table 2: Detection results(%) of baseline fully-supervised models on SODA10M labeled set.

| Model | Split | mAP | Pedestrian | Cyclist | Car | Truck | Tram | Tricycle | Params |
|---|---|---|---|---|---|---|---|---|---|
| RetinaNet [35] 1x | Val | 32.7 | 23.9 | 37.3 | 55.7 | 40.0 | 36.6 | 3.0 | **36.4M** |
| RetinaNet [35] 2x | Val | 35.0 | 26.6 | 39.4 | 57.2 | 41.8 | 38.2 | 6.5 | **36.4M** |
| RetinaNet [35] 2x | Test | 34.0 | 24.9 | 36.9 | 57.5 | 44.7 | 32.1 | 7.8 | **36.4M** |
| Faster RCNN [44] 1x | Val | 37.9 | 31.0 | 43.2 | 58.3 | 43.2 | 41.3 | 10.5 | 41.4M |
| Faster RCNN [44] 2x | Val | 38.7 | 32.5 | 43.6 | 58.9 | 43.7 | 40.8 | 12.6 | 41.4M |
| Faster RCNN [44] 2x | Test | 36.7 | 29.5 | 40.1 | 59.7 | 47.2 | 32.3 | 11.7 | 41.4M |
| Cascade RCNN [2] 1x | Val | **41.9** | 34.6 | 46.7 | 61.9 | 47.2 | 45.1 | 16.0 | 69.2M |
| Cascade RCNN [2] 1x | Test | **39.4** | 31.9 | 43.4 | 62.6 | 50.0 | 36.8 | 11.9 | 69.2M |

### 4.3 Self-Supervised Learning Benchmark

Self-supervised learning, especially contrastive learning methods, has raised attraction recently as it learns effective transferable representations via pretext tasks without semantic annotations. Traditional self-supervised algorithms [10, 41, 16] are usually pre-trained on ImageNet [7], while there is no available self-supervised benchmark tailored for autonomous driving. Therefore, we evaluate the performance of existing mainstream self-supervised methods when pre-trained on autonomous driving datasets, including SODA10M, BDD100K [60], nuScenes [1] and Waymo [48], and provide some interesting observations. Due to the limit of hardware resources, we only use a 5-million unlabeled subset in each experiment by default.

#### 4.3.1 Datasets Comparison

We utilize different self-supervised methods to pre-train on four different datasets (SODA10M, BDD100K [60], nuScenes [1] and Waymo [48]), and then report the fine-tuning performance on different downstream tasks (object detection task on BDD100K [60] and SODA10M, semantic segmentation task on Cityscapes [6] and BDD100K [60], instance segmentation task on Cityscape).

Table 3: Comparison of downsteam tasks' performance with different upstream pre-training datasets. IS, SS, OD stand for instance segmentation, semantic segmentation and object detection task respectively.

| Pre-trained Dataset | Method | Cityscape (IS) | BDD100K (SS) | Cityscape (SS) | BDD100K (OD) | SODA10M (OD) |
|---|---|---|---|---|---|---|
| nuScenes [1] | MoCo-v1 [21] | 31.4 | 57.0 | 73.6 | 31.1 | 36.2 |
| | MoCo-v2 [5] | 31.5 | 56.8 | 73.8 | 30.9 | 36.8 |
| Waymo [48] | MoCo-v1 [21] | 31.4 | 57.0 | 73.8 | 31.2 | 37.1 |
| | MoCo-v2 [5] | 31.8 | 56.6 | 73.5 | 31.1 | 37.1 |
| BDD100K [60] | MoCo-v1 [21] | 31.8 | 57.9 | 74.5 | 31.4 | 37.1 |
| | MoCo-v2 [5] | 32.0 | 57.5 | **74.4** | 31.3 | 37.8 |
| SODA10M | MoCo-v1 [21] | **33.9** | **59.3** | **75.2** | **31.5** | **38.9** |
| | MoCo-v2 [5] | **33.7** | **58.2** | 74.2 | **31.4** | **38.7** |

As shown in Table 3, **SODA10M outperforms other autonomous driving datasets in 9 out of 10 downstream tasks, which verifies that the superior scale and diversity of SODA10M can bring better generalization ability.** Besides, we found that **diversity matters more than scale for the upstream pre-training dataset.** For example, even with the scale 10 times smaller than nuScenes [1] and Waymo [48], BDD100K [60] achieves the better fine-tuning performance compared with nuScenes [1] and Waymo [48] due to its relatively larger diversity (*i.e.*, BDD100K [60] covers more scenes like snowy and longer driving time of 1111 hrs). To conclude, collected from 32 cities, with driving time 25x, 5000x, 4300x longer and dataset scale 100x, 10x and 10x larger than the BDD100K [60], nuScenes [1] and Waymo [48], SODA10M can better serve as a promising pre-training dataset for different self-supervised learning methods.

On the other hand, experiments result in Table 4 show that the model pre-trained on SODA10M performs equivalent or slightly worse than the one on ImageNet [7]. Global contrastive learning methods (*i.e.*, MoCo-v1 [21], MoCo-v2 [5]), which take each image as a class, may not be suitable for SODA10M with multiple instances in one image. Besides, dense pixel-wise contrastive learning method (*i.e.*, DenseCL [54]) also fails to deal with the images of complex driving scene. **We think new contrastive loss, which is specially designed for the images with multiple instances and complex driving scenes, should be proposed to boost the self-supervised performance when pre-training on these autonomous driving datasets.** Then we can truly take advantage of pre-training on the images in autonomous driving domain and achieve the best performance of fine-tuning different tasks in the same domain. We leave the design of multi-instance-based contrastive loss for future work since it is out of the scope of this paper.

### 4.3.2 Methods Comparison

As shown in Table 4, **part of the global contrastive methods (including MoCo-v1 [21], MoCo-v2 [5]) and dense contrastive method (DenseCL [54]) can achieve better results when pre-training on SODA10M, while the other methods perform worse than ImageNet fully supervised pre-training.** We also observe that dense contrastive method (DenseCL [54]) shows excellent result when pre-training on ImageNet [7] (39.9% compared with 39.0% of MoCo-v1 [21]), but relatively poor on SODA10M unlabeled set (38.1% compared with 38.9% of MoCo-v1 [21]) due to the reason that pixel-wise contrastive loss may not suitable for complex driving scenarios. Besides, by comparing the result of method with † with the original method, we found that more pre-training iterations can bring better performance (*i.e.*, SimCLR [4]† exceed SimCLR [4] by an average margin of 1.2% over four downstream tasks). Comparisons on 2x schedule are illustrated in Appendix C.

Table 4: Fine-tuning results(%) of self-supervised models evaluated on SODA10M labeled set (SODA), Cityscapes [6] and BDD100K [60]. mIOU(CS), mIOU(BDD) denote for semantic segmentation performance on Cityscapes and BDD100K respectively. † represents for training with additional 5-million data. FCN-16s is a modified FCN with stride 16 used in MoCo [21]. 1x and 90k denote fine-tuning 12 epochs and 90k iterations.

| Pre-trained Dataset | Method | Faster-RCNN 1x (SODA) | | | RetinaNet 1x (SODA) | | | FCN-16s 90k | |
| | | mAP | AP50 | AP75 | mAP | AP50 | AP75 | mIOU (CS) | mIOU (BDD) |
|---|---|---|---|---|---|---|---|---|---|
| | random init | 23.0 | 40.0 | 23.9 | 11.8 | 20.8 | 12.0 | 65.3 | 50.7 |
| | super. IN | 37.9 | 61.6 | 40.4 | 32.7 | 53.9 | 33.9 | 74.6 | 58.8 |
| ImageNet [7] | MoCo-v1 [21] | 39.0 | 62.0 | 41.6 | 33.8 | 54.9 | 35.2 | 75.3 | 59.7 |
| | MoCo-v2 [5] | 39.5 | 62.7 | 42.4 | 35.2 | 56.4 | 36.8 | 75.7 | 60.0 |
| | SimCLR [4] | 37.0 | 60.0 | 39.4 | 29.0 | 49.0 | 29.3 | 75.0 | 59.2 |
| | SwAV [3] | 35.7 | 59.9 | 36.9 | 26.4 | 45.7 | 26.3 | 73.0 | 57.1 |
| | DetCo [56] | 38.7 | 61.8 | 41.3 | 33.3 | 54.7 | 34.3 | **76.5** | **61.6** |
| | DenseCL [54] | **39.9** | 63.2 | 42.6 | **35.7** | 57.3 | 37.2 | 75.6 | 59.3 |
| SODA10M | MoCo-v1 [21] | 38.9 | 62.1 | 41.2 | 33.4 | 54.4 | 34.6 | 75.2 | 59.3 |
| | MoCo-v1† [21] | **39.0** | 62.6 | 41.9 | **33.8** | 55.2 | 35.2 | **75.5** | **59.5** |
| | MoCo-v2 [5] | 38.7 | 61.5 | 41.4 | 33.3 | 54.1 | 34.7 | 74.2 | 58.2 |
| | MoCo-v2† [5] | 38.6 | 61.3 | 41.4 | 33.2 | 54.6 | 34.6 | 74.5 | 58.9 |
| | SimCLR [4] | 35.9 | 59.5 | 37.4 | 28.7 | 48.7 | 29.1 | 73.3 | 57.3 |
| | SimCLR† [4] | 37.1 | 60.9 | 39.8 | 30.5 | 51.3 | 31.2 | 73.5 | 58.8 |
| | SwAV [3] | 33.4 | 57.1 | 34.5 | 24.5 | 43.2 | 24.6 | 68.6 | 54.2 |
| | DetCo [56] | 37.7 | 60.6 | 40.1 | 32.4 | 54.1 | 33.4 | 74.1 | 59.3 |
| | DenseCL [54] | 38.1 | 60.8 | 40.5 | 33.6 | 54.8 | 35.0 | 75.2 | 57.4 |

### 4.3.3 Video-based Self-supervised Methods

**SODA10M can also serve for evaluating different video-based self-supervised methods due to the reason that it contains detailed timing information.** To generate videos, we transform the unlabeled set into video frames with an interval of 10 seconds. Based on these 90k generated videos, traditional video-based self-supervised methods consider frames in same video as positive samples, frames in other videos as negative samples and perform similar contrastive loss.

Experiment results of different video-based self-supervised methods are summarized in Table 5. Equipped with MLP projection head and more data augmentations, MoCo-v2 [5] (28.9%, 74.4%

on SODA10M and Cityscape) achieves better performance on most downstream tasks than MoCo-v1 [21] (27.9%, 73.6%) and VINCE [18] (27.6%, 72.6%). With the stronger data augmentation method jigsaw, VINCE [18] performs better and achieves similar performance to MoCo-v2 [5] (with an average difference of 0.05% over 4 different downstream tasks).

Table 5: Fine-tuning results(%) of video-based self-supervised models on SODA10M labeled set (SODA), Cityscapes [6] and BDD100K [60]. mIOU(CS), mIOU(BDD) denote for semantic segmentation performance on Cityscapes and BDD100K respectively. All models are pre-trained on SODA10M unlabeled set.

| | Faster-RCNN 1x (SODA) | | | RetinaNet 1x (SODA) | | | FCN-16s 90k | |
|---|---|---|---|---|---|---|---|---|
| Method | mAP | AP50 | AP75 | mAP | AP50 | AP75 | mIOU (CS) | mIOU (BDD) |
| Video MoCo-v1 [21] | 34.9 | 57.8 | 36.6 | 27.9 | 47.3 | 28.2 | 73.6 | 57.3 |
| Video MoCo-v2 [5] | 34.8 | 57.0 | 36.5 | **28.9** | 48.6 | 29.5 | **74.4** | 56.8 |
| Video VINCE [18] | 34.9 | 57.7 | 36.9 | 27.6 | 47.1 | 28.0 | 72.6 | **57.4** |
| Video VINCE+Jigsaw [18] | **35.5** | 58.1 | 37.0 | 28.2 | 48.1 | 28.6 | 74.1 | 56.9 |

## 4.4 Semi-Supervised Learning Benchmark

Semi-supervised learning has also attracted much attention because of its effectiveness in utilizing unlabeled data. We compare the naive pseudo labeling method with present state-of-the-art consistency-based methods for object detection (*i.e.*, STAC [46] and Unbiased Teacher [38]) on 1-million unlabeled images considering the time limit. For pseudo labeling, we first train a supervised Faster-RCNN [44] model on the training set with the ResNet-50 [23] backbone for 12 epochs. Then we predict bounding box results on the images of unlabeled set, bounding boxes with predicted score larger than 0.5 are selected as the ground-truth label to further train a new object detector.

We report the experiment results of different semi-supervised methods in Table 6 and observe that all semi-supervised methods exceed the results of using only labeled data. As for pseudo labeling, adding an appropriate amount of unlabeled data (50K to 100K) brings a greater improvement, but continuing to add unlabeled data (100K to 500K) results in a 1.4% decrease due to the larger noise. On the other hand, **consistency-based methods combined with pseudo labeling outperform pseudo labeling by a large margin**. STAC [46] exceeds pseudo labeling by 2.9%, and Unbiased Teacher [38] continues to improve by 3.4% due to the combination of exponential moving average (EMA) and focal loss [35].

Table 6: Detection results(%) of semi-supervised models on SODA10M dataset. Pseudo labeling (50K), Pseudo labeling (100K) and pseudo labeling (500K) mean using 50K, 100K and 500K unlabeled images, respectively.

| Model | mAP | AP50 | AP75 | Pedestrian | Cyclist | Car | Truck | Tram | Tricycle |
|---|---|---|---|---|---|---|---|---|---|
| Supervised | 37.9 | 61.6 | 40.4 | 31.0 | 43.2 | 58.3 | 43.2 | 41.3 | 10.5 |
| Pseudo Labeling (50K) | $39.3^{+1.4}$ | 61.9 | 42.4 | 32.6 | 44.3 | 60.4 | 43.8 | 42.4 | 12.1 |
| Pseudo Labeling (100K) | $39.9^{+2.0}$ | 62.7 | 42.6 | 33.1 | 45.2 | 60.7 | 44.8 | 43.3 | 12.1 |
| Pseudo Labeling (500K) | $38.5^{+0.6}$ | 61.0 | 41.3 | 32.1 | 43.4 | 59.6 | 42.6 | 42.2 | 11.0 |
| STAC [46] | $42.8^{+4.9}$ | 64.8 | 46.0 | 35.7 | 46.4 | 63.4 | 47.5 | 44.4 | 19.6 |
| Unbiased Teacher [38] | $\mathbf{46.2}^{+8.3}$ | 70.1 | 50.2 | 33.8 | 50.2 | 67.9 | 53.9 | 55.2 | 16.4 |

## 4.5 Discussion

We directly compare the performance of state-of-the-art self/semi-supervised methods on SODA10M with supervised Faster-RCNN [44] in Table 7. In this table, we illustrate the overall performance (mAP) for daytime/night domain and car detection results of 18 fine-grained domains (considering different periods, locations and weather conditions).

### 4.5.1 Domain Adaptation Results

From Table 7, we observe that there exists a considerable gap between the domain of daytime and night. Since the supervised method is only trained on the data during the daytime, the gap between

Table 7: Domain adaptation results(%) of self/semi-supervised methods in different domains on SODA10M dataset. '-' means no validation image in this domain.

| Model | Overall mAP | City street (Car) | | | Highway (Car) | | | Country road (Car) | |
|---|---|---|---|---|---|---|---|---|---|
| | | Clear | Overcast | Rainy | Clear | Overcast | Rainy | Clear | Overcast |
| Daytime | | | | | | | | | |
| Supervised | 43.1 | 70.0 | 64.9 | 56.6 | 68.3 | 65.9 | 65.9 | 69.4 | 63.5 |
| MoCo-v1 [21] ImageNet | $44.2^{+1.1}$ | 71.5 | 65.8 | 56.9 | 69.0 | 66.8 | 67.3 | 72.0 | 66.0 |
| MoCo-v1 [21] SODA10M | $43.8^{+0.7}$ | 71.3 | 66.0 | 55.8 | 69.4 | 67.4 | 68.0 | 72.8 | 65.5 |
| STAC [46] | $\underline{45.3}^{+2.2}$ | 74.2 | 69.6 | 58.0 | 71.7 | 70.3 | 70.7 | 75.2 | 69.8 |
| Unbiased Teacher [38] | $\mathbf{47.7}^{+4.6}$ | 73.0 | 68.1 | 55.3 | 69.1 | 62.0 | 71.3 | 72.6 | 70.0 |
| Night | | | | | | | | | |
| Supervised | 21.1 | 36.3 | 37.7 | - | 37.5 | 37.3 | 79.5 | 38.9 | 72.8 |
| MoCo-v1 [21] ImageNet | $22.0^{+0.9}$ | 39.5 | 43.4 | - | 41.7 | 41.5 | 80.6 | 42.5 | 73.2 |
| MoCo-v1 [21] SODA10M | $22.7^{+1.6}$ | 41.6 | 46.2 | - | 42.1 | 41.8 | 79.8 | 45.4 | 74.1 |
| STAC [46] | $\underline{28.2}^{+7.1}$ | 45.5 | 46.8 | - | 46.2 | 45.6 | 83.7 | 47.2 | 75.4 |
| Unbiased Teacher [38] | $\mathbf{39.7}^{+18.6}$ | 65.3 | 66.2 | - | 66.2 | 67.2 | 83.6 | 67.5 | 75.2 |

day (43.1%) and night (21.1%) is particularly obvious. By adding diverse unlabeled data into training, **the semi-supervised methods show a more significant improvement in the night domain than the self-supervised methods** (+18.6% of Unbiased Teacher [38] vs. +1.6% of MoCo-v1 [21]).

For self-supervised learning, pre-training on ImageNet brings almost equal improvement in both day and night domain (+1.1% vs. +0.9%), and we assume that is because ImageNet [7] contains common images which achieve no special helps to the performance of night domain. On the other hand, **pre-training on SDOA10M, which contains massive images collected at night, brings double improvement in night domain (+1.6%) compared to day domain (+0.7%).**

### 4.5.2 Performance and Training Efficiency Comparison

Although trained with a smaller set of unlabeled data (1-million vs. 5-million), consistency-based semi-supervised methods work much better than all self-supervised methods under same labeled data either from the aspect of overall performance (46.2% vs. 38.9% for Unbiased teacher [38] and MoCo-v1 [21] respectively, as shown in Table 6 and Table 4) or the total training time (2.8×8 GPU days vs. 8.4×8 GPU days, as shown in Appendix B). Better performance will be achieved when combining self-supervised and semi-supervised methods.

## 5 Conclusion

Focusing on self-supervised and semi-supervised learning, we present SODA10M, a large-scale 2D autonomous driving dataset that provides a large amount of unlabeled data and a small set of high-quality labeled data collected from various cities under diverse weather conditions, periods and location scenes. Compared with the existing self-driving datasets, SODA10M is the largest in scale and obtained in much more diversity. Furthermore, we build a benchmark for self-supervised and semi-supervised learning in autonomous driving scenarios and show that SODA10M can serve as a promising dataset for training and evaluating different self/semi-supervised learning methods. Inspired by the experiment results, we summarize some guidance for dealing with SODA10M dataset. For self-supervised learning, method which focuses on dealing with multi-instance consistency should be proposed for driving scenarios. For semi-supervised learning, domain adaptation can be one of the most important topics. For both self and semi-supervised learning, efficient training with high-resolution and large-scale images will be promising for future research. We hope that SODA10M can promote the exploration and standardized evaluation of advanced techniques for robust and real-world autonomous driving systems.

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
