# OpenReview forum: "SODA10M: A Large-Scale 2D Self/Semi-Supervised Object Detection Dataset for Autonomous Driving"
_NeurIPS.cc/2021/Track/Datasets_and_Benchmarks/Round2 — NeurIPS 2021 Datasets and Benchmarks Track (Round 2)_

### Official Review · Reviewer_xKdJ · 2021-09-16
**SODA10M: A Large-Scale 2D Self/Semi-Supervised Object Detection Dataset for Autonomous Driving**

**Rating:** 6
**Confidence:** 5
**Correctness:** the evaluation methods and experiment…
**Clarity:** The paper is easy to understand and w…

**Strengths:**

The paper is well written, and easy to understand.

**Weaknesses:**

1. As described in the paper, the images are capured every 10 seconds per frame, it is hard to understand why. The consecutiveness among different frames is really low under such setting, and it is easy to capture high frame rate images.  As described in line 125, "They have to use the mobile phone or driving recorder (1080P+) to obtain images every 10 seconds per frame" , why "have to", the mobile phone and  driving recorder are easy to capture video, this is really hard to understand.

2. In the released dataset contains 10 million images and 20K labeled images. However, it is unclear that how the number of labeled images affects the final results in self/semi-supervied object detection task. In line 165, the paper says that "we believe that SODA10M provides sufficient labeled samples to evaluate different self/semi-supervised learning methods," how to evaluate whether the labeled images are sufficient in the self/semi-supervised learning methods?

3. In section 4.3.3, how does it work in video based self-supervised methods? The images are captured every 10s, though time information exists, the images are in-continuous.

4. The task of each table should be clearly given, which results of task are reported in Table.5?



**Additional Feedback:**

see weakness.

**Documentation:**

Sufficient details on data collection and organization are provided.

**Relation To Prior Work:**

The differences between the work and previous methods same clear.

**Summary And Contributions:**

The paper releases a Large-Scale 14 2D Self/semi-supervised Object Detection dataset for Autonomous driving, named as SODA10M, which contains 10 million unlabeled images and 20K images labeled with 6 representative object categories. To improve diversity, the images are collected within 27833 driving hours under different weather conditions, periods and location scenes of 32 different cities. We provide extensive experiments and deep analyses of existing popular self-supervised and semi-supervised approaches, and give some interesting findings in autonomous driving scope.

---

> ### Author Response · Authors · 2021-09-27
> **Reply to Reviewer4**
>
> Thank you for the comments. All comments are summarized and addressed as follows.
>
> ****Q1: It is hard to understand why the images are captured every 10 seconds per frame.****
>
> R1: We found collecting frames with a 10s interval in a video is a good trade-off between retaining data diversity and using smaller storage space. If the time interval is smaller, the difference between two adjacent frames may become smaller and it will cover fewer scenes (with the same number of images), which may reduce the overall diversity of the dataset. From our experiments, we observe that the diversity of the data has a relatively large impact on the final performance (refer to line 242-245). If the time interval is larger, both crowdsourced people (taxi drivers) and users in academic field will be difficult to validate their techniques considering that the total storage space of SODA10M with 10s interval is already 2T.
>
> ****Q2: It is unclear that how the number of labeled images affects the final results and how to evaluate whether the labeled images are sufficient to evaluate self/semi-supervised methods?****
>
> R2: To show that how the number of labeled images affects the final results in self/semi-supervised object detection task, we further conduct the experiments to show the self-supervised (with MoCov1) object detection performance under various downstream SODA10M dataset sizes (20\%, 50\%, 100\%) and upstream pre-training datasets (Waymo, SODA10M). The results are shown in the following:
>
> |      |20%| 50%|100%|
> |:---------- | :-------------| :-------------- | :-----------------|
> | Pre-train on Waymo | 26.6 | 33.2 | 37.1 |
> | Pre-train on SODA10M | 29.2 (+2.6)|35.3 (+2.1)| 38.9 (+1.8)|
>
> Observation can be made that more downstream labeled data bridge the gap between different self-supervised models. On the other hand, too few labeled images tend to have a large standard deviation (SD) of the final performance, e.g, pre-training on SODA10M receives an 0.75 SD on 20\% SODA10M labeled set, compared with the 0.11 SD on 100\% SODA10M labeled set.
> Detailed experiment results are added to Appendix C.
>
> We show that our SODA10M labeled set is sufficient to evaluate different self/semi-supervised learning methods by comparing the number of labeled images with Pascal VOC (20K vs. 21K).
> As Pascal VOC is considered as a popularly used downstream dataset for the object detection task to evaluate different self/semi-supervised learning methods, it's reasonable to believe that SODA10M also contains enough labeled images to evaluate the existing methods.
>
> ****Q3: In section 4.3.3, how does it work in video-based self-supervised methods? The images are captured every 10s, though time information exists, the images are in-continuous.****
>
> R3: Thanks for your valuable comments. Exploring the temporal and continuous information among framework is indeed a worthy and interesting direction for video-based self-supervised learning. Since there are currently rare self-supervised works investigating how to use temporal information, our Table 5 focuses on validating whether treating more frames of the same video as positive samples under MOCO and VINCE framework can improve the performance, which may encode implicit temporal clues. In future, we believe this dataset will serve as a valuable benchmark for coming researches on temporal-based self-supervised learning and we also try our best to incorporate such new video techniques into this benchmark.
>
> ****Q4: The downstream task of Table 5 should be clearly given.****
>
> R4: Keeping the same as Table 4, Table 5 reports the fine-tuning results of different video self-supervised models evaluated on SODA10M labeled dataset(object detection), Cityscape(semantic segmentation) and BDD100K(semantic segmentation). Thanks for your remind and the details are also added to the caption of Table 5 in the revised version.

---

### Official Review · Reviewer_88A8 · 2021-09-20
**Good paper, with beneficial large and diverse dataset.**

**Rating:** 7
**Confidence:** 4

**Strengths:**

This dataset is among the top largest datasets to be made available to the general public. The diversity of the data collected is extremely important to build any robust downstream perception algorithm.

The fact that they have blurred out possible identification data in the dataset is very good and shows a clear thought towards ethical implications of the dataset.

This dataset has been used as part of an ICCV challenge and hence is being widely used by many teams, hence showcasing the quick and far reach of this dataset.

The domain adaptation intent is good.

The authors address the questions posed from round 1 of the submissions.

**Weaknesses:**

The paper needs to address the impact of the low frame rate. At highways speeds, 10 secs is a long time. The dataset would not work too well due to the reason mentioned above for any kind of tracking algorithm development and benchmarks. Though that doesn't seem to be the main intent of the authors, it would be of super huge value to have data at a higher frame rate for tracking purposes.

The fact that annotations are only done on 6 different categories, does not address the concern from the other datasets highlighted in line 12 of the abstract.

In automotive datasets, from personal experience and other works, we have noticed that position and pose of the camera in data collections can have a big impact on the performance of the models. Especially the difference between these across the different vehicles. The authors have neither mentioned about that in the paper nor the impact of characteristics such as height of the camera from the ground, pitch, yaw on the performance of the algorithms. Also information on the kind of vehicles used would be helpful. In large scale crowdsourced datasets, this is an incredibly tough ask to maintain consistency across all vehicles and it would help to strengthen the paper to mention some information on this.

**Additional Feedback:**

Overall the paper is good and thanks to the authors to bring this out to the community in general for improvement of self and semi supervised algorithms in the Autonomous driving space.

**Clarity:**

The paper is mostly well-written with a consistent messaging and clear progression. There are some grammatical and spelling errors to be fixed, and with that taken care of along with addressing some of the other weaknesses, this paper is a top 25% among accepted papers for me. They do use a long appendix to add information to the paper.

**Correctness:**

The dataset has mostly been collected in a sound way with keen observation to various noise/varying factors in the environment. The diversity of the data is a key contribution.

**Documentation:**

The link to the dataset was provided. They go into a lot of details, but my earlier mentioned weaknesses along with some below mentioned points, once addressed would help with providing an exhaustive information. Some more information on the instructions given to the drivers, the start and stop of the data collections, the vehicle speeds during this time would be immensely helpful for many other applications.

**Ethics:**

They have shared information including their privacy policy for the data. A clear focus on the ethical aspects is showcased especially with the example of blurring out identifiable information from the images. This is a very good practice.

The question 5 of the checklist was answered as N/A even though the data was indeed crowdsourced. Some more insight into the instructions given to the taxi drivers/installation entity will help, not just from an ethics stand point but also from the viewpoint of the data collection.

**Relation To Prior Work:**

Yes.

**Summary And Contributions:**

The paper presents a dataset collected in China highlighting the scale and diversity of the data collection. With 10M unlabeled images and 20K labeled images, this dataset focuses on the impact towards self and semi- supervised object detection, semantic segmentation, instance segmentation pipelines. The authors also share details about some experiments conducted to compare this dataset with other top datasets in the field. The comparison is effective and the impact is showcased with the results from various top models and techniques.

---

> ### Author Response · Authors · 2021-09-27
> **Reply to Reviewer3**
>
> Thank you for your comments. All comments are summarized and addressed as follows.
>
> ****Q1: The dataset with 10s interval of adjacent images would not work too well for tracking algorithm development and benchmarks.****
>
> R1: We found collecting frames with a 10s interval in a video is a good trade-off between retaining data diversity and using smaller storage space. If the time interval is smaller, the difference between two adjacent frames may become smaller and it will cover fewer scenes (with the same number of images), which may reduce the overall diversity of the dataset. From our experiments, we observe that the diversity of the data has a relatively large impact on the final performance (refer to line 242-245). If the time interval is larger, both crowdsourced people (taxi drivers) and users in academic field will be difficult to validate their techniques considering that the total storage space of SODA10M with 10s interval is already 2T. Due to the above limits, the tracking task is not covered in SODA10M dataset by now.
>
> ****Q2: The fact that annotations are only done on 6 different categories, does not address the concern from the other datasets highlighted in line 12 of the abstract.****
>
> R2: We believe these 6 necessary categories used for annotating already cover the most common instances in the driving scene, and are the most important classes for object detection tasks in autonomous driving.
> Besides, we note that well-known datasets like KITTI only have 3 classes (Car/Pedestrian/Cyclist) and Waymo also only contains 3 classes (Vehicle/Pedestrian/Cyclist).
> Although nuScenes has more categories, most categories are covered by SODA10M, e.g., adult, child, construction worker and police officer in nuScenes are indicated as Pedestrian in SODA10M.
>
> To clarify, the "limited domains" mentioned in line 12 of the abstract denotes the different scenarios (e.g., different weather, locations, cities) of unlabeled images since the purpose of building SODA10M is to exploit large-scale unlabeled data.
>
> ****Q3: The authors have neither mentioned about the impact of different vehicles nor  characteristics such as position and pose of the camera from the ground, pitch, yaw on the performance of the algorithms. Information on the kind of vehicles or the camera settings used would be helpful.****
>
> R3: An detailed instruction is given to each taxi driver to maintain consistency across different characteristics of the crowdsourced images.
> Thus we don't mention the impact of different vehicles or camera settings in this paper.
> Some important settings are listed as follows:
> 1) The kind of the vehicles used is passenger car, for example, sedan or van;
> 2) The height of the camera is in the range of 1.4m-1.5m from the ground, depending on the different crowdsourced vehicles;
> 3) The camera should be mounted on the rear mirror in the car, facing straight ahead.
> Besides, horizon needs to be kept at the center of the image, and the occlusion inside the car should not exceed 15\% of the whole picture;
> 4) The detailed camera settings, including distortion, exposure, white balance, image resolution, etc, are introduced;
> 5) Quality control measures are adopted in the pre and post-collection.
>
> The details are added to the data collection part (Section 3.1) in revised version and the full participants’ instructions can be found in Appendix A.
>
> ****Q4: Question 5 of the checklist should be answered and some more insight into the instructions given to the taxi drivers will help.****
>
> R4: Thanks for your remind, we have filled the details in question 5 of the checklist in the revised version. Besides, the full participants’ instructions can be found in Appendix A.

---

### Official Review · Reviewer_URPg · 2021-09-21
**Release of 2K Labelled Data and 10 Million Raw Data**

**Rating:** 4
**Confidence:** 5

**Strengths:**

The paper proposed a good idea about using 2K labelled data and 10M unlabeled data. However, after studying the paper and reading the benchmark section of figure 3, it is certainly very underwhelmed. And more details needs to be studied and included to make a strong argument.

**Weaknesses:**

* The author claims scale as a one of the unique contributions, as compared to the release of Waymo(with annotations), SODA10M contains 10 million data frames(without annotations). I do not think it is a fair comparison, as the AV industry does not lack raw data, but annotated data.
* There are some experimental analyses of self-supervised/semi supervised approaches included in the paper, but there is no novel contribution from this paper by using these existing toolsets and code implementations.


**Additional Feedback:**

NA

**Clarity:**

* The paper repeatedly mentions too many times about its contribution to releasing a dataset.
* I would recommend to cut down some over-emphasized common knowledge, and develop more about the unique contribution of this paper.


**Correctness:**

Since it is a release of datasets, I hope that more sound details about data collection and annotation have been conducted. Specifically,compared to talking about common knowledge such as collecting data with different weather locations for generalization purposes, discussing how to design the driving fleet across different cities to collect data in the most effective way could also be very interesting points.

**Documentation:**

Documentation is under-developed in terms of how exactly the data collection process has been designed, or how to incorporate more feedback to improve the existing labor intensive data collection process.

**Relation To Prior Work:**

Discussions about emerging methods from semi-supervised and self-supervised learning areas are covered.

**Summary And Contributions:**

Raw imaginary data has been collected across 32 cities with a total of 27833 driving hours, which must consume lots of time, investment for these collective efforts.

---

> ### Author Response · Authors · 2021-09-27
> **Reply to Reviewer2**
>
> Thanks for your comments and questions about SODA10M. All comments are summarized and addressed as follows.
>
> ****Q1: Unfair comparison between Waymo(with annotations) and SODA10M (which contains 10 million data frames without annotations). Besides, the AV industry does not lack raw data, but annotated data.****
>
> R1: We would like to emphasize here that this paper focuses on building a self/semi-supervised benchmark which shall focus more on the global diversity in the whole dataset (labeled+unlabeled) rather than the annotated number. In addition to the scale aspect, SODA10M is also better than Waymo in the number of collected cities (32 vs. 3), driving hours (27833 vs. 6.4) and scenes (as in Table 10 of Appendix D).
> Moreover, although the industry does not lack raw data, there is currently no available standard resources and benchmark which provides such large-scale raw data (which is non-trivial) for the research community to develop more advanced self/semi-supervised learning techniques.
>
> ****Q2: There are some experimental analyses of self-supervised/semi-supervised approaches included in the paper, but there is no novel contribution from this paper by using these existing toolsets and code implementations.****
>
> R2: The NeurIPS Dataset and Benchmark track guidelines suggest that it is a novel venue for publishing highly valuable machine learning datasets/benchmarks and dataset development, instead of new methods. The main contribution of this paper is to propose a new large-scale dataset aiming at self/semi-supervised learning and provide the benchmark results of existing methods on this dataset while providing detailed dataset collections.
>
> Besides, we provide intriguing findings and insights under autonomous driving scope for future research on this research field, e.g., 1) larger scale and diversity of the pre-training dataset can bring better generalization ability in downstream tasks (refer to Section 4.3.1); 2) novel contrastive methods for autonomous driving should be proposed to handle images with multiple instances and complex driving scenes (refer to Section 4.3.1); 3) the semi-supervised methods show a more significant improvement in the night domain than the self-supervised methods, and pre-training on SDOA10M, which contains massive images collected at night, brings double improvement in night domain compared to day domain (refer to Section 4.5.1).
>
> ****Q3: Documentation of data collection process is under-developed.****
>
> R3: Thanks for your great advice. More data collection details (e.g., sensor specifications, quality control and data distribution) are added to Section 3.1 in the revised version. Besides, the full instructions for participants can be found in Appendix A.

---

### Official Review · Reviewer_uMGY · 2021-09-23
**Useful dataset for a variety of tasks in autonomous driving with appropriate and well motivated experimental validation**

**Rating:** 7
**Confidence:** 4
**Clarity:** Yes, the paper is clear, well written…

**Strengths:**

- SODA contains 10 million unlabeled images where 20K images are labeled with 6 object categories. The images are collected from 27,833 driving hours under varying weather conditions and time-of-day periods from 32 different cities for the task of object detection. This is a comparatively large dataset and is suitable for pre-training and for self/semi-supervision methods on downstream tasks.
- Problem statement is well motivated.
- Quantitatively better performance on downstream tasks from self-supervised training methods with the proposed dataset as compared with relevant baseline datasets.
- Sound reasoning for the clarification presented in L158-166 regarding the small number of annotated images available in the dataset.
- Necessary ethical clarifications and privacy compliance have been addressed in L138-143 and Appendix A.
- Supplementary material includes implementation details and also includes improvements from previous versions.
- Self-supervised learning benchmarks in Section 4.3 are conclusive and demonstrate the applicability of the proposed dataset on several relevant tasks including video-based methods. The conclusions drawn are useful insights.
- Diversity statistics of the images presented in Tables 6,7 and 8 in the Supplementary material help verify the claim presented in L219-224 regarding improved diversity of scenes as a factor in improved performance.


**Weaknesses:**

- The dataset compilation procedure poses questions on quality control and how crowdsourcing data from volunteer/paid taxi drivers can be uniform. Are the performance gains in the experiments solely based on increased dataset size? Are there any quality control measures in place?
- Results when pre-trained with ImageNet outperform SODA10M for several tasks including the segmentation performance presented in the Supplementary material. The authors should explain this clearly.

**Additional Feedback:**

L70: preforms -> performs

**Correctness:**

Yes, the claims made in the submission seem correct and the dataset is constructed in a somewhat sound manner (some reservations). The evaluation benchmarks are exhaustive and performed correctly to demonstrate the utility of the proposed dataset in several tasks. The limitations are mentioned in the Supplementary material.

**Documentation:**

SODA10M dataset was made available for the ICCV2021 SSLAD challenge. The documentation is sufficient and the detail on data collection, statistics, and maintenance have been suitably discussed and addressed. Relevant code to reproduce some benchmark experiments has been made available.

**Ethics:**

No, there are no ethical concerns that warrant further review. Necessary clarifications have been presented in L138-143 and Appendix A. Terms of use and licenses are included in the supplementary material.

**Relation To Prior Work:**

Yes, previous works and relevant datasets have been discussed in the context of the autonomous vehicle vision niche. Further, supervised and self-supervised learning methods have been discussed in light of their progress relevant to baseline experiments.

**Summary And Contributions:**

- SODA contains 10 million unlabeled images where 20K images are labeled with 6 object categories from 32 different cities.
- The authors have suitably identified a good application for the large-scale dataset without labels, and discuss the scale, diversity, and generalization.
- Dataset collected by crowdsourcing from taxi drivers following the data protection laws by blurring personal information (faces, license plates, etc) and/or omitting GPS coordinates etc. The (small set of) images are exhaustively annotated following a three-step process.
- The dataset has been released and was also part of the CCV2021 SSLAD challenge.

---

> ### Author Response · Authors · 2021-09-27
> **Reply to Reviewer1**
>
> We appreciate for your comments. All comments are summarized and addressed as follows.
>
> ****Q1: The dataset compilation procedure poses questions on quality control and how crowdsourcing data from volunteer/paid taxi drivers can be uniform.****
>
> R1: We conduct the pre-collection and post-collection quality control to guarantee the high quality of the SODA10M dataset. The pre-collection quality control includes checking the camera position and imaging quality on each taxi. The post-collection quality control includes manual verification and those images of low quality (unclear imaging, strong reflection and incorrect camera position) will be returned for rectification.
> The details are added to the data collection part (Section 3.1) in revised version.
>
> To keep the images uniform from crowdsourcing system, we gave instruction for each taxi driver, showing them the detailed camera location and settings, including distortion, exposure, white balance, image resolution, etc.
> The full instructions for participants can be found in Appendix A.
>
> ****Q2: Are the performance gains in the experiments solely based on increased dataset size?****
>
> R2: No, the performance gains in the experiments are based on both quantity and diversity.
> For the quantity part, as shown in Table 4, training with 5M more images (result with $\dagger$) brings better performance. For example, SimCLR† exceeds SimCLR by an average margin of 1.2\% over four downstream tasks.
> On the other hand, diversity is another important aspect. As shown in line 242-245, even with the scale 10 times smaller than nuScenes and Waymo, BDD100K achieves the better fine-tuning performance compared with nuScenes and Waymo due to its relatively larger diversity (i.e., BDD100K covers more scenes like snowy and longer driving time of 1111 hrs).
>
> ****Q3: Explain the results when pre-trained with ImageNet outperform SODA10M for several tasks including the segmentation performance presented in the Supplementary material.****
>
> R3: We speculate the reason why the ImageNet pretraining models perform better than the SODA10M models is that current self-supervised learning methods cannot fully leverage the potential representation power of images in complex scenarios, e.g., self-driving scenes in the SODA10M dataset.
> As we explained in line 250-253. Global contrastive learning methods (i.e., MoCo-v1, MoCo-v2), which take each image as a class, may not be suitable for SODA10M with multiple instances in one image. Besides, dense pixel-wise contrastive learning method (i.e., DenseCL) also fails to deal with the images of complex driving scenes.

---

### Author Response · Authors · 2021-09-27
**Updates on the revised version**

Based on the reviewer's comments, here we provide the summary about the updates of the revised version, note that all updates are highlighted by orange color:

1. More data collection details (e.g., sensor specifications, quality control and data distribution) are added to Section 3.1. (R1, R2, R3)

2. Experiments on how the number of labeled images affects the final fine-tuning performance are added to the Appendix C. (R4)

3. Question 5 of the checklist is filled. (R3)

4. The detailed fine-tuning tasks’ information is given in Table 5. (R4)

5. Fix some grammatical and spelling errors. (R1, R3)

---

### Decision · Program_Chairs · 2021-10-10

**Decision:**

Accept

**Comment:**

The reviews are mostly positive. Although the dataset has a small number of labels, it can be a useful resource for research on self-supervised learning. The diversity of coverage is definitely a big plus. I am thus happy to accept i.